# CD8+ tissue-resident memory T cells induce oral lichen planus erosion via cytokine network

**Maofeng Qing[1,2†], Dan Yang[1†], Qianhui Shang[1], Jiakuan Peng[1], Jiaxin Deng[1], Jiang Lu[1], Jing Li[1], HongXia Dan[1], Yu Zhou[1*], Hao Xu[1*], Qianming Chen[3*]**

[1]State Key Laboratory of Oral Diseases, National Clinical Research Center for Oral Diseases, Research Unit of Oral Carcinogenesis and Management, Chinese Academy of Medical Sciences, West China Hospital of Stomatology, Sichuan University, Chengdu, China; [2]Department of Stomatology, The Second Affiliated Hospital of Chongqing Medical University, Chongqing, China; [3]Key Laboratory of Oral Biomedical Research of Zhejiang Province, Affiliated Stomatology Hospital, Zhejiang University School of Stomatology, Hangzhou, China

**Abstract** CD8+ tissue-resident memory T (CD8+ Trm) cells play key roles in many immune-inflammation-related diseases. However, their characteristics in the pathological process of oral lichen planus (OLP) remains unclear. Therefore, we investigated the function of CD8+ Trm cells in the process of OLP. By using single-cell RNA sequencing profiling and spatial transcriptomics, we revealed that CD8+ Trm cells were predominantly located in the lamina propria adjacent to the basement membrane and were significantly increased in patients with erosive oral lichen planus (EOLP) compared to those with non-erosive oral lichen planus (NEOLP). Furthermore, these cells displayed enhanced cytokine production, including IFN-γ (Interferon-gamma, a pro-inflammatory signaling molecule), TNF-α (Tumor Necrosis Factor-alpha, a cytokine regulating inflammation), and IL-17 (Interleukin-17, a cytokine involved in immune response modulation), in patients with EOLP. And our clinical cohort of 1-year follow-up was also supported the above results in RNA level and protein level. In conclusion, our study provided a novel molecular mechanism for triggering OLP erosion by CD8+ Trm cells to secrete multiple cytokines, and new insight into the pathological development of OLP.

**\*For correspondence:**
812471898@qq.com (YZ);
hao.xu@scu.edu.cn (HX);
qmchen@scu.edu.cn (QC)

†These authors contributed equally to this work

**Competing interest:** The authors declare that no competing interests exist.

## Editor's evaluation

Overall, this manuscript is a valuable contribution to the field of human immunology and provides solid data and interpretations. Very little is known about oral lichen planus so this dataset may also serve as a public resource. Thank you for your contribution on mapping the cellular landscape of this poorly understood condition.

## Introduction

Oral lichen planus (OLP) is a chronic inflammatory disease on the oral mucosa of unknown etiology mediated by T cells with a 1.01% global prevalence rate (*González-Moles et al., 2021*; *Jiang et al., 2022b*). Its clinical manifestations are varied and chronic non-healing (*Radwan-Oczko, 2013*). Especially, recurrent erosion or ulcers in patients with erosive oral lichen planus (EOLP) were mostly accompanied by pain and discomfort, which worsen when eating or speaking, and adversely affect the physical and mental health of patients (*Rotaru et al., 2020*). Notably, EOLP has a significantly higher risk of malignant transformation than non-erosive oral lichen planus (NEOLP) (*Arduino et al., 2021*).

To reduce the psychological and economic burden of OLP patients, improve their quality of life, and decrease the risk of cancer, it is crucial to maintain the disease in a relatively stable non-erosive stage for as long as possible. However, clinical experience suggests that OLP often exhibits a prolonged and recurrent disease course, with alternating periods of non-erosive and erosive lesions. Despite this, the underlying causes and mechanisms of lesion type switching remain unclear (*Gorouhi et al., 2014*).

The pathological feature of OLP is characterized by dense infiltration of T cells in the lamina propria, disruption of the basement membrane, and degeneration of basal keratinocytes (*Xu et al., 2022*). Previous studies have shown that OLP patients have immune function disorders both locally and systemically, with the infiltration of CD8[+] T cells in the OLP lamina propria near the basement membrane (*Neppelberg et al., 2001*). CD8[+] T cells are related to the liquefaction of basal cells, and it is believed that the immune microenvironment of OLP lesions has changed, especially in CD8[+] T cells, which produce a complex network of cytokines and chemokines, such as IFN-γ, TNF-α, IL1α, and IL17, among others (*Firth et al., 2015*; *Ke et al., 2017*; *Piccinni et al., 2014*; *Shaker and Hassan, 2012*; *Viguier et al., 2015*).

Tissue-resident memory T (Trm) cells are a recently described population of terminally differentiated T cells, which are crucial for local immunity (*Mueller et al., 2014*). Trm cells are mainly present in various barrier tissues, where CD8[+] Trm cells can persist locally for a long time in the absence of relevant antigens (*Szabo et al., 2019*). Recent studies had revealed that CD8[+] Trm cells play important roles in the occurrence and development of many chronic inflammatory diseases such as psoriasis and vitiligo (*Eberle et al., 2016*; *Richmond et al., 2018*; *Watanabe, 2019*).

CD8[+] Trm cells are considerable components of local immunity, yet their presence, distribution, and function in OLP are poorly understood. This study aims to investigate the presence and spatial distribution of CD8[+] Trm cells in different clinical manifestations of OLP, and to determine their functional role, especially in the context of the heterogeneity observed between NEOLP and EOLP. Additionally, the study aims to explore the underlying molecular mechanisms that contribute to the development of erosive lesions in OLP.

## Results

### Single-cell RNA sequencing revealed the cell composition of OLP with different clinical subtypes

To investigate the cellular composition and comprehensive transcriptional effects of OLP, we performed single-cell RNA sequencing (scRNA-seq) of NEOLP (NEOLP, *n* = 3) and EOLP (EOLP, *n* = 2). The final dataset comprised 46,377 cells, with an average of 1743 genes per cell. Visualization using uniform manifold approximation and projection (UMAP) revealed 47 distinct cell clusters (*Figure 1A*) that were annotated as 8 major cell types (*Figure 1B*). Although there was no significant difference in T cell proportion between NEOLP and EOLP, T cell was still the major cell proportion in EOLP. Surprisingly, although T cells, the hallmark cells of OLP, constituted the highest proportion of all samples, the proportion of cells in EOLP-2 was the lowest at 47.47%, while the proportion of T cells in the NEOLP-3 case is the highest at 71.93%. The cell composition analysis also revealed an increase in the proportion of B cells and myeloid cells in EOLP (*Figure 1C, D*). Our analysis of myeloid cells revealed some interesting findings. We observed that neutrophils were significantly more abundant in EOLP than NEOLP (*Figure 1—figure supplement 1A, B*). Similarly, plasmacytoid (pDC) were also more common in EOLP compared to NEOLP. However, NEOLP had a higher number of myeloid (mDC) than EOLP (*Figure 1—figure supplement 1A–D*). Upon conducting GO analysis, we found that neutrophil cells in EOLP were significantly enriched in defense response and positive regulation of cytokine production (*Figure 1—figure supplement 1E*).

### The cell proportion of CD8[+] Trm cells was increased in EOLP

The typical pathological manifestation of OLP is band infiltration of T cells in the lamina propria. At present, most views believe that OLP is an immune-related disease mediated by T cells (*Feldmeyer et al., 2020*). So, our next further analyzed the T cell population.

Since NK and T cells are also derived from lymphoid progenitor cells, NK and T cells are developmentally closer and mature NK cells express CD3 (a T cell signature gene) after activation, while the cytotoxic response of mature NK cells is similar to that of CTL (*Abel et al., 2018*). Therefore, in the

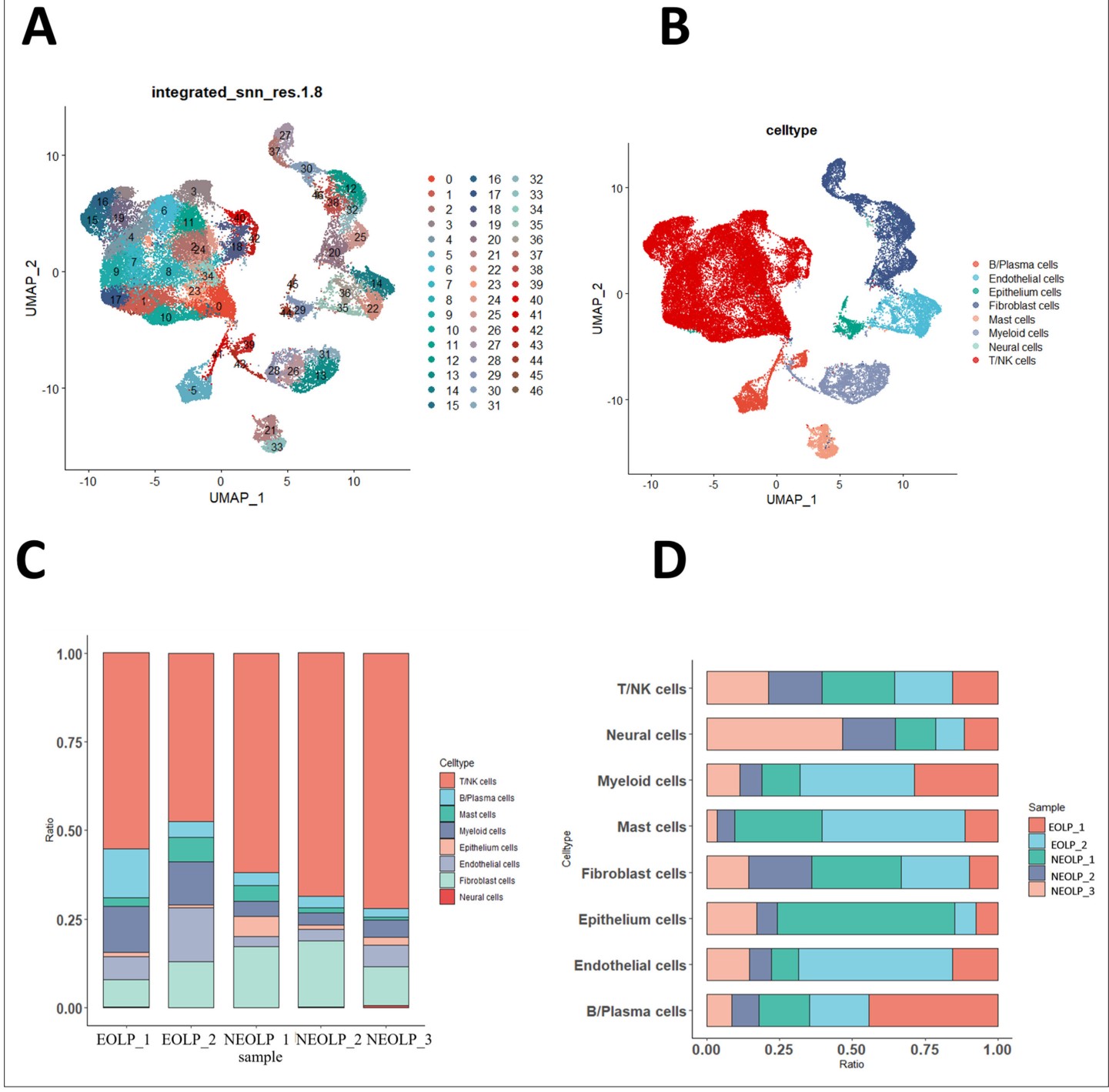

**Figure 1.** Oral lichen planus (OLP) tissue transcription atlas. (**A**) Uniform manifold approximation and projection (UMAP) plot of 46,377 cells colored by cell type (erosive oral lichen planus, EOLP) (*n* = 2) and non-erosive oral lichen planus (NEOLP) (*n* = 3). (**B**) Eight major cell types were identified by UMAP plot. (**C**) Bar plot showing the percentage of cell subsets in different samples. (**D**) The proportion of each sample in different cell subsets.

The online version of this article includes the following source data, source code, and figure supplement(s) for figure 1:

**Source code 1.** UMAP visualization and cell typing of OLP including identification of eight major cell types.

**Source data 1.** Percentage of eight major cell types in each OLP sample.

**Source data 2.** The proportion of each OLP sample in the eight main cell types.

**Figure supplement 1.** The characteristics of myeloid cells in oral lichen planus (OLPs).

**Figure supplement 1—source data 1.** The percentage of myelid subsets in different OLP samples.

*Figure 1 continued on next page*

*Figure 1 continued*

**Figure supplement 1—source data 2.** Percentage of each cell subset of myeloid in EOLP and NEOLP.

**Figure supplement 1—source code 1.** UMAP visualization and cell typing of myeloid cells in OLP samples, along with GO enrichment analysis of differentially expressed myeloid cell genes between EOLP and NEOLP.

initial study, T cells and NK cells were included together to form separate NK and T cell populations, and the NK/T population constituted the majority of cells sequenced in the OLP in the study data. Subsequently, by further clustering, the genes of the NK/T cell population, the NK/T cell population was divided into a total of 23 cell subgroups (*Figure 2—figure supplement 1A, B*). After further annotation of the NK/T cell subsets with marker genes, six large cell subpopulations were described: NK cells, CD8$^+$ T cells, CD4$^+$ T cells, CD4$^+$ CD8$^+$ T cells, CD8$^+$ Trm cells, and CD4$^+$ CD8$^+$ Trm cells (*Figure 2A, B*).

We observed remarkable differences in the proportion of T cell populations between patients with different clinical types of OLP. EOLP exhibited fewer CD8$^+$ T cells but more CD4$^+$ T cells, CD8$^+$ Trm cells, and CD4$^+$ CD8$^+$ T cells than NEOLP. The proportion of CD4$^+$ CD8$^+$ Trm and NK cells was similar in both types (*Figure 2C*).

## The immune function of CD8$^+$ Trm cells was activated in EOLP

To further explore the function of T cells, this study demonstrated that in the CD4$^+$ T cells subset, LTB, an inflammatory response gene, was significantly active, while CCL4, CCL4L2, and CLL5 that guide immune cells to migrate to inflammatory sites were activated in CD8$^+$ Trm and CD8$^+$ T cells subset, and the expression of the latter is more obvious; the gene BTG1, which inhibits cell proliferation, is significantly expressed in the CD4$^+$CD8$^+$T and CD4$^+$CD8$^+$TRM subgroups (*Figure 2D* and *Figure 2— figure supplement 1C*).

Our study also explored the production of pro-inflammatory cytokines by different T cell subsets. We observed that the CD8$^+$ T cell subset displayed significantly higher expression of GZMK, a gene known to contribute to the cytotoxicity and induction of apoptosis in target cells, compared to other subsets. Moreover, CD8$^+$ Trm cells in particular exhibited higher expression levels of GZMA, GZMK, TNF, PRF1, and other genes associated with inflammatory factors, when compared to other subgroups. It may have contributed to the worsening of the clinical manifestations of OLP (*Figure 2E*).

Further analysis of the significantly different genes between NEOLP and EOLP indicated that B2M, closely related to the stability of MHC class I molecules and antigen presentation, and HIF1A, which regulates immune cell function and inflammation, were found to be significantly expressed in EOLP. Moreover, the EOLP expressed a number of markers involved in the differentiation and function of Trm cells, including CD69, IL7R, CD7, FOS, etc. (*Figure 2—figure supplement 2A*, *Supplementary file 1a*).

To investigate the difference between CD8$^+$ Trm cells in EOLP and NEOLP, we performed differential expression analysis and found that the CD8$^+$Trm marker gene CD69, GNLY which can play a cytotoxic role, and multiple pro-inflammatory factor-related genes, such as GZMB, IFNG, TNF, and PRF1, were significantly increased in the CD8$^+$ Trm subgroup in EOLP. These findings suggest that the CD8$^+$ Trm activity in EOLP may be significantly enhanced than NEOLP (*Figure 2—figure supplement 2B*, *Supplementary file 1b*).

GO enrichment analysis indicated that CD8$^+$ Trm differentially expressed genes (DEGs) in EOLP between NEOLP were more significantly enriched in pathways such as positive regulation of cytokine production, regulation of T cell activation, and positive regulation of cell activation (*Figure 2F*). It shows that CD8$^+$ Trm subsets in EOLP have a different state and expressions compared with NEOLP subsets, and may have a more active state and a greater ability to produce cytokine. The pseudotime analysis also suggests that CD8$^+$ Trm cells may represent one of the terminal states of T cell differentiation and development in OLP (*Figure 2G*), which may have a profound relationship with their clinical manifestations.

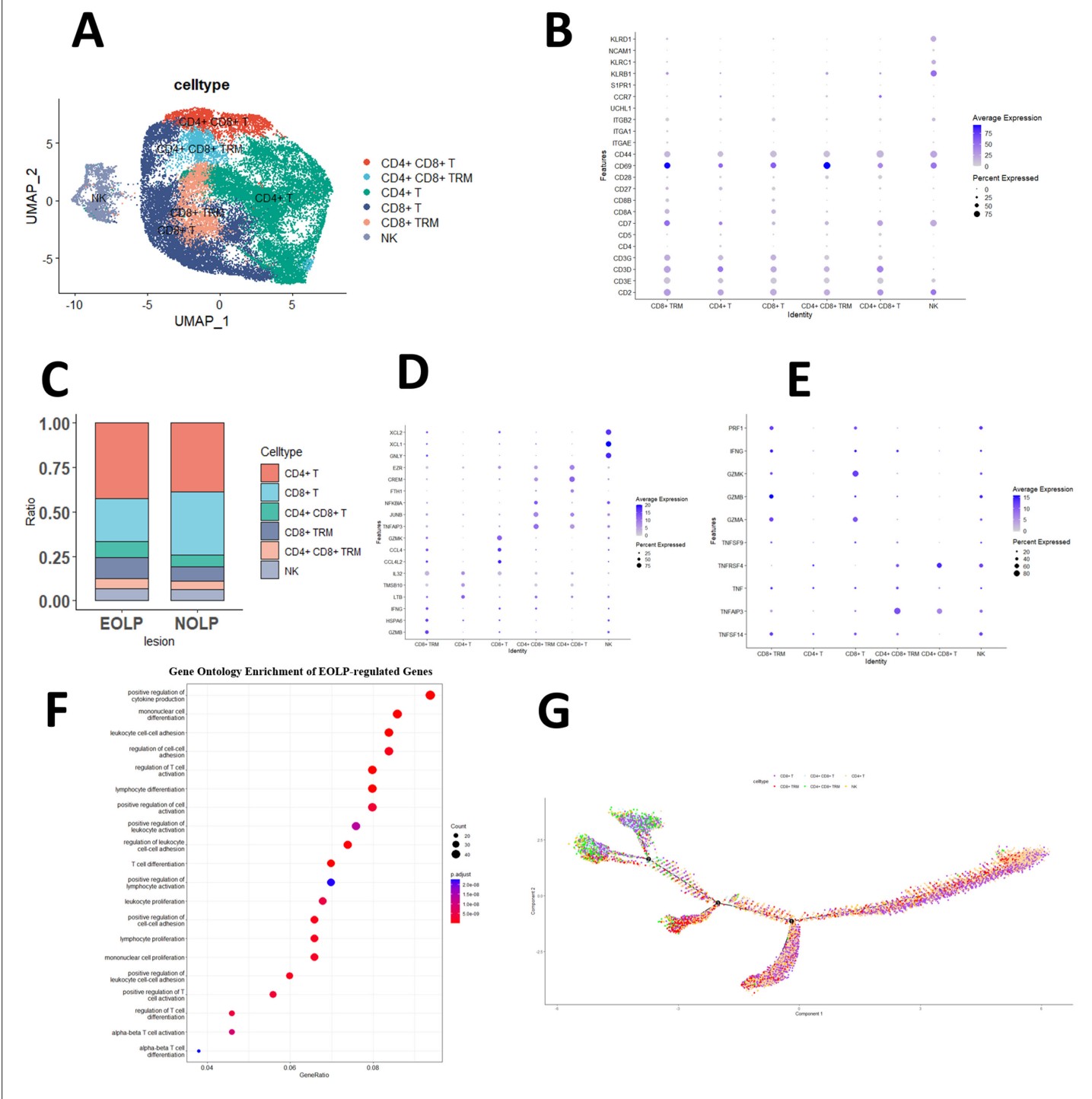

**Figure 2.** CD8[+] Trm in oral lichen planus (OLP) patients have different transcriptomic landscapes in different clinical presentations. (**A**) Uniform manifold approximation and projection (UMAP) plot of NK/T cells colored by subtype. (**B**) Dot plot of marker genes in NK/T cells subsets. (**C**) Proportion of NK/T cell subsets in OLP with different clinical manifestations. (**D**) Dot plot of significant differentially expressed genes (DEGs) in NK/T cells subsets. (**E**) Dot plot of differential expression of inflammatory cytokine product genes in NK/T cell subsets. (**F**) Dot plot of GO enrichment analysis of DEGs between erosive oral lichen planus (EOLP) and non-erosive oral lichen planus (NEOLP). (**G**) Pseudotime trajectory of NA/T cells in OLP.

The online version of this article includes the following source data, source code, and figure supplement(s) for figure 2:

**Source code 1.** Comprehensive analysis of NK/T cell in OLP samples: UMAP visualization, cell typing, marker genes, DEGs, inflammatory cytokine profiles, GO enrichment analysis, and pseudotime trajectory between EOLP and NEOLP.

*Figure 2 continued on next page*

*Figure 2 continued*

**Source data 1.** Percentage of NK/T cell subsets in EOLP and NEOLP.

**Figure supplement 1.** Transcriptional Profiling of NK/T Cells in OLP.

**Figure supplement 1—source code 1.** Transcriptional Profiling of NK/T Cells in OLP: Clustering, DEGs, and Cell Subpopulation Expression.

**Figure supplement 2.** Analysis of differentially expressed genes (DEGs) in oral lichen planus (OLPs).

## Spatial transcriptomics revealed CD8+ Trm cells were adjacent to the epithelium of OLP and its products may induce epithelial erosion and promote the development of disease

Cell types, relative positions among cells, and the levels of gene expression of cell populations together determine their function in biological tissues. To investigate the spatial heterogeneity between normal oral mucosa and OLP, we performed the spatial transcriptomics (ST) analysis of normal oral mucosa (*n* = 2), NEOLP (*n* = 3), and EOLP (*n* = 1).

The tissues in this study covered spots ranging from the lowest 296 spots in the NEOLP-1 to the highest 965 spots in the NEOLP-3, while the per-sample capture factors ranged from 2869 to 3884 (*Supplementary file 1c*). First, we used keratin to characterize the basal layer of the epithelium. The normal oral mucosa and NEOLP had intact epithelium, while some NEOLP samples showed atrophied and thinned epithelium, while others showed hyperplasia and thickening. In EOLP, it can be seen that the epithelium on the left side of the tissue is intact, while the epithelium on the right side is missing (*Figure 3A*).

After co-localization with the marker gene of CD8+ Trm cells, it was found that CD8+ Trm cells were more distributed in EOLP than NEOLP, and more distributed in NEOLP than normal oral mucosa. And it can be observed that whether in normal oral mucosa, NEOLP or EOLP, CD8+ Trm cells are mostly distributed adjacent to the epithelium, while in EOLP tissues, CD8+ Trm cells in the areas where the epithelium is lost are correspondingly reduced, and there are CD8+ Trm cells in the deeper lamina propria of the corresponding tissues (*Figure 3B*).

We observed that the expression of TNF, IL17A/IL17RA, IFNGR1, etc. was higher than in normal oral mucosa. And the signals of the effector receptor IFNGR1 of IFN and the effector receptor IL17RA of IL17 were significantly enhanced in EOLP compared with NEOLP.

## Cohort studies confirmed the core genes of CD8+ Trm cells were closely associated with the erosion and process of OLP

To objectively illustrate the molecular mechanism of OLP erosion, we established a clinical cohort with 40 participants, which were confirmed by clinical manifestation and pathological diagnosis. The basic information of the clinical cohort is shown in *Table 1*. This study included 40 OLP patients, there were 15 females and 12 males in the NEOLP group and 8 females and 5 males in the EOLP group. The mean age of the NEOLP group was 30.07 ± 11.34 years and that of the EOLP group was 45.08 ± 12.86 years. There were seven smoking patients and nine drinking patients in NEOLP and two smoking patients and three drinking patients in EOLP. The mean VAS score of the NEOLP group was 1.41 ± 0.89 and that of the EOLP group was 2.46 ± 1.391. The mean course of disease was 14.85 ± 27.932 months in the NEOLP group and 14.85 ± 27.932 months in the EOLP group. There was no significant difference in clinical information between the two groups (*Table 1*).

Through differential gene analysis, it was found that in NEOLP and EOLP, the expression of multiple core factors and effector cytokines or their receptor transcription factors of CD8+ Trm cells, including ITGA1, LITAF, SKIL, etc., and related immune factor receptors such as TNFRSF6B, IL17RA, and IFNAR1 were significantly increased in the EOLP than in the NEOLP (*Figure 4A, B*; *Table 2*).

To further investigate the clinical relevance of these findings, we divided the clinical cohort into two groups based on the 1-year follow-up: the recurrent erosion (RE) group (the interval between erosions <3 months) and the persistent non-erosion (PNE) group (≥3 months without any form of erosion). In addition to the above clinical factors being included in the two groups for statistical comparison, the diagnosis (NEOLP or EOLP) and medication (divided into three groups: local glucocorticoids, local glucocorticoids + immunosuppressant, and other drugs) were analyzed (*Table 3*), in which glucocorticoids were all used for local and external use. Differences in clinical information other than diagnosis

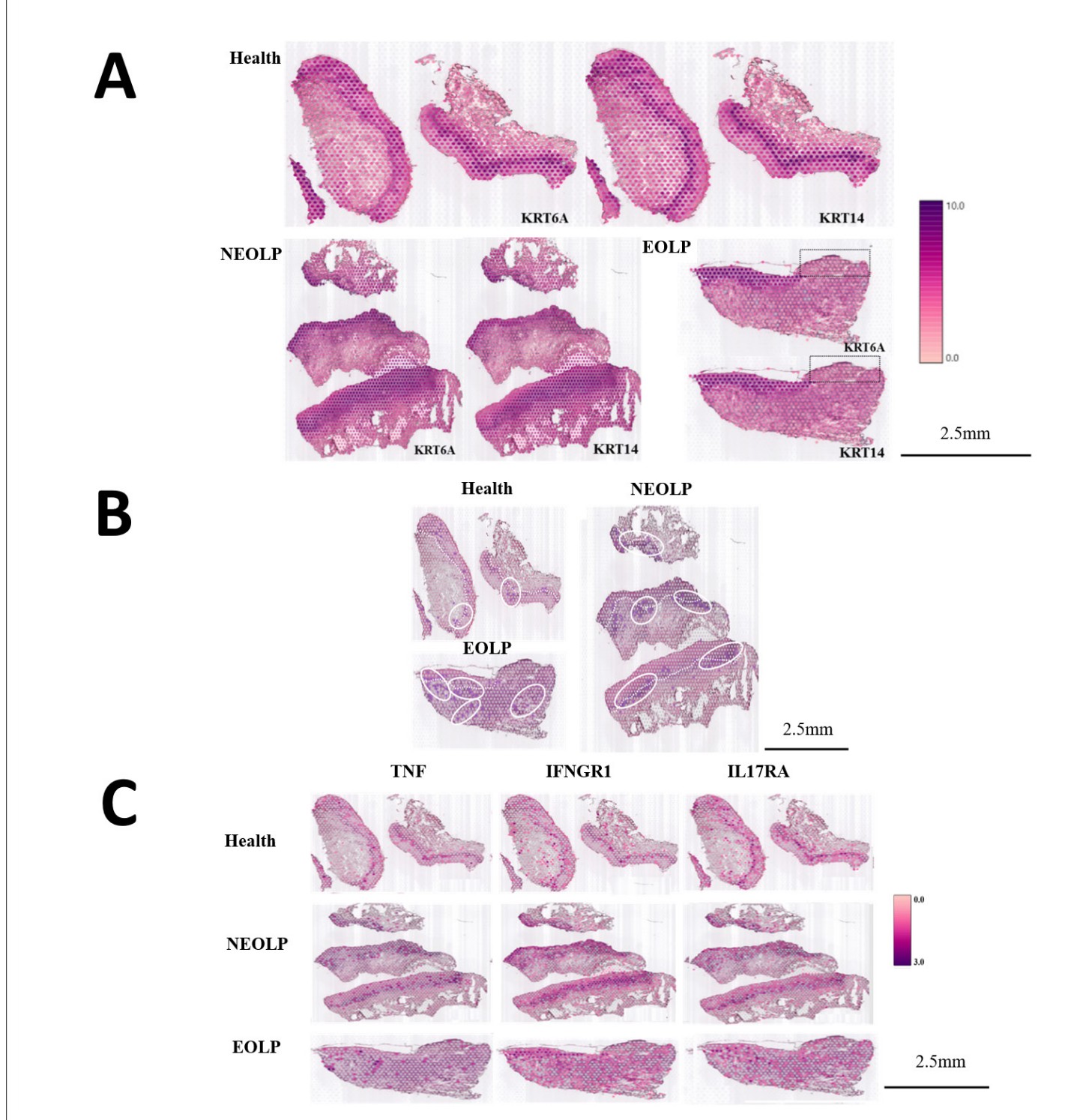

**Figure 3.** CD8+ Trm in oral lichen planus (OLP) patients have different transcriptomic landscapes in different clinical presentations. (**A**) Marked the epithelial layer area of each group of samples. (**B**) CD8+ Trm distribution area. The dark purple spots are the CD8+ TRM marker gene co-expression spots; the area in the white circle is the CD8+ TRM distribution area; the white dotted line is the basement membrane. (**C**) TNF/IFNGR1/IL17RA distribution area.

**Table 1.** Analysis of clinical variables in different clinical types of oral lichen planus (OLP).

| Factors | NEOLP | EOLP | $t/c^2$ | p |
|---|---|---|---|---|
| Gender | | | 0.129 | 0.72 |
| Female | 15 | 8 | | |
| Male | 12 | 5 | | |
| Age (years) | 39.07 ± 11.34 | 45.08 ± 12.86 | −1.502 | 0.348 |
| Smoking | | | 0.559 | 0.455 |
| Yes | 7 | 2 | | |
| No | 20 | 11 | | |
| Drinking | | | 0.44 | 0.507 |
| Yes | 9 | 3 | | |
| No | 18 | 10 | | |
| VAS scoring | 1.41 ± 0.89 | 2.46 ± 1.391 | −2.91 | 0.341 |
| Course of disease (months) | 14.85 ± 27.932 | 23.38 ± 43.065 | −0.755 | 0.12 |

were still not statistically significant for the clinical outcome of OLP. Moreover, the multiple core factors and effector cytokines of CD8[+] Trm cells were found to be consistently upregulated in the recurrent erosion group, further supporting the role of CD8[+] Trm cells in OLP progression (*Figure 4C, D*).

To mitigate confounding factors and minimize systematic bias, we employed multivariate logistic regression to analyze patients with different clinical types and outcomes of OLP. The multivariate logistic model was used to perform regression analysis on the effects of CD8[+] Trm-related factors on the clinical manifestations and clinical outcomes of OLP after adjusting for clinical factors such as age, gender, smoking, and drinking. We found that the core genes of CD8[+] Trm cells are closely related to the clinical manifestations of OLP, and the expression differences of marker genes such as ITGA1, LITAF, SKIL, and cytokine-related genes such as IL17RA, IFNG41, IFNAR1, and TNFRSF6B were statistically significant between the two clinical types. The core genes of CD8[+] Trm cells are also closely related to the erosion of OLP. Marker genes such as CD69, ITGA1, LITAF, SKIL, and cytokine-related genes such as IL17RA, IFNG41, IFNAR1, and GZMB are associated with two clinical outcomes of persistent non-erosion and recurrent erosion. The differences in expression were statistically significant and were significantly increased in the recurrent erosion group (*Table 2*).

## CD8[+] Trm cells may participate in the erosion of OLP by secreting active cytokines

Immunofluorescence experiments verified that there were CD8[+] Trm cells in NEOLP and EOLP. The content of CD8[+] Trm cells in EOLP was significantly increased than that in NEOLP. It can be seen from the location that CD8[+] Trm cells are mostly close to the basal layer. In NEOLP, the basement membrane was basically intact, and sporadic T cells entered the epithelial layer, while in EOLP, CD8[+] Trm cells accumulated more obviously under the erosive epithelium, and the number was obviously higher than in NEOLP, and the mucosal epithelial basement membrane was not clear in some places, more T cells were entering the mucosal epithelial layer, some of which were CD8[+] Trm cells (*Figure 5A–D*).

According to the classic distinguishing and defining characteristics of memory T cell subsets, a CD8[+] Trm sorting strategy was developed (*Figure 4E*). The OLP tissue cell suspension was incubated with antibodies for sorting, and it was found that the content of CD8[+] Trm cells in EOLP was significantly higher than that in NEOLP (p < 0.05) (*Figure 5F, G*). We further verified the previous results in scRNA-seq and ST and found that the difference in CD8[+]Trm content in different clinical types of OLP is strongly correlated with the clinical feature, which may be one of the reasons for the different clinical manifestations.

To investigate the functional differences between CD8[+] Trm cells in EOLP and NEOLP, we stimulated sterile-sorted CD8[+] Trm cells from both tissue types with phytohemagglutinin (PHA) and performed an enzyme-linked immunosorbent assay (ELISA) to detect cytokine production. Our results

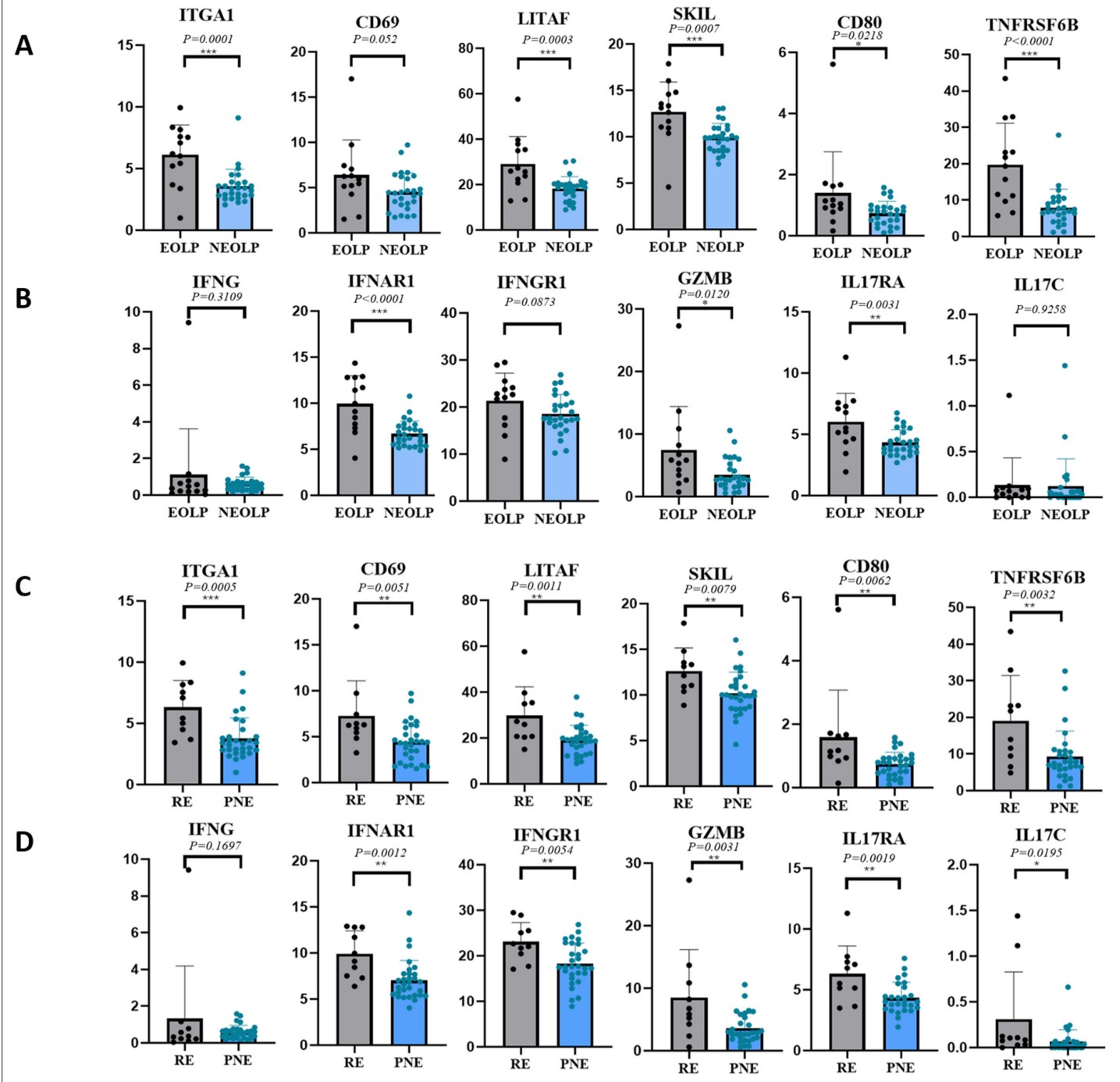

**Figure 4.** CD8[+] Trm in oral lichen planus (OLP) patients have different transcriptomic expression in different clinical presentations and outcomes. (**A, B**) CD8[+] Trm marker genes and related inflammatory factor genes in erosive oral lichen planus (EOLP)/non-erosive oral lichen planus (NEOLP). (**C, D**) CD8[+] Trm marker genes and related inflammatory factor genes in recurrent erosion (RE)/persistent non-erosion (PNE).

The online version of this article includes the following source data for figure 4:

**Source data 1.** Expression of CD8+ Trm marker genes and related inflammatory factor genes in EOLP and NEOLP in clinical cohort.

**Source data 2.** Expression of CD8+ Trm marker genes and related inflammatory factor genes in RE and PNE in clinical cohort.

**Table 2.** Logistic regression analysis of CD8$^+$ Trm core transcriptome in different oral lichen planus (OLP) clinical types and clinical outcomes.

| | OLP clinical types | | OLP clinical outcomes | |
|---|---|---|---|---|
| Gene | p | OR (95% CI) | p | OR (95% CI) |
| ITGA1 | 0.006* | 2.009 (1.221, 3.307) | 0.017* | 1.814 (1.114, 2.954) |
| LITAF | 0.011* | 1.197 (1.041, 1/376) | 0.026* | 1.169 (1.019, 1.342) |
| CD69 | 0.057 | 1.337 (0.991, 1.804) | 0.026* | 1.846 (1.076, 3.168) |
| SKIL | 0.006* | 1.868 (1.197, 2.914) | 0.046* | 1.482 (1.006, 2.182) |
| GZMB | 0.05 | 1.281 (1.000, 1.641) | 0.041* | 1.370 (1.012, 1.854) |
| IL17RA | 0.025* | 1.943 (1.087, 3.471) | 0.036* | 3.270 (1.084, 9.867) |
| IFNG | 0.237 | 1.467 (0.777, 2.768) | 0.412 | 1.392 (0.632, 3.067) |
| IFNGR1 | 0.086 | 1.156 (0.980, 1.365) | 0.015* | 1.386 (1.067, 1.801) |
| IFNAR1 | 0.005* | 1.987 (1.237, 3.191) | 0.022* | 1.601 (1.069, 2.399) |
| TNFRSF6B | 0.011* | 1.221 (1.047, 1.425) | 0.05 | 1.104 (1.000, 1.218) |
| RUNX1 | 0.009* | 1.703 (1.139, 2.545) | 0.07 | 1.283 (0.980, 1.682) |
| IL23A | 0.395 | 1.730 (0.489, 6.119) | 0.015* | 9.971 (1.555, 63.950) |

*p < 0.05; **p < 0.01; ***p < 0.001.

**Table 3.** Analysis of clinical variables in different clinical outcomes of oral lichen planus (OLP).

| Factors | Persistent non-erosion | Recurrent erosion | t/c² | p |
|---|---|---|---|---|
| Gender | | | 0.034 | 0.853 |
| Female | 17 | 6 | | |
| Male | 13 | 4 | | |
| Age (years) | 39.07 ± 12.71 | 45.00 ± 9.09 | −1.214 | 0.161 |
| Smoking | | | 1.195 | 0.274 |
| Yes | 8 | 1 | | |
| No | 22 | 9 | | |
| Drinking | | | 0 | 1 |
| Yes | 9 | 3 | | |
| No | 21 | 7 | | |
| VAS scoring | 1.57 ± 0.97 | 2.30 ± 1.57 | −1.76 | 0.213 |
| Course of disease (months) | 17.63 ± 32.90 | 17.60 ± 36.20 | 0.003 | 0.965 |
| Diagnosis | | | 8.547 | 0.003* |
| NEOLP | 24 | 3 | | |
| EOLP | 6 | 7 | | |
| Medication | | | 5 | 0.082 |
| Glucocorticoids | 15 | 1 | | |
| Glucocorticoids + immunosuppressant | 10 | 6 | | |
| Other drugs | 5 | 3 | | |

*p < 0.05; **p < 0.01; ***p < 0.001.

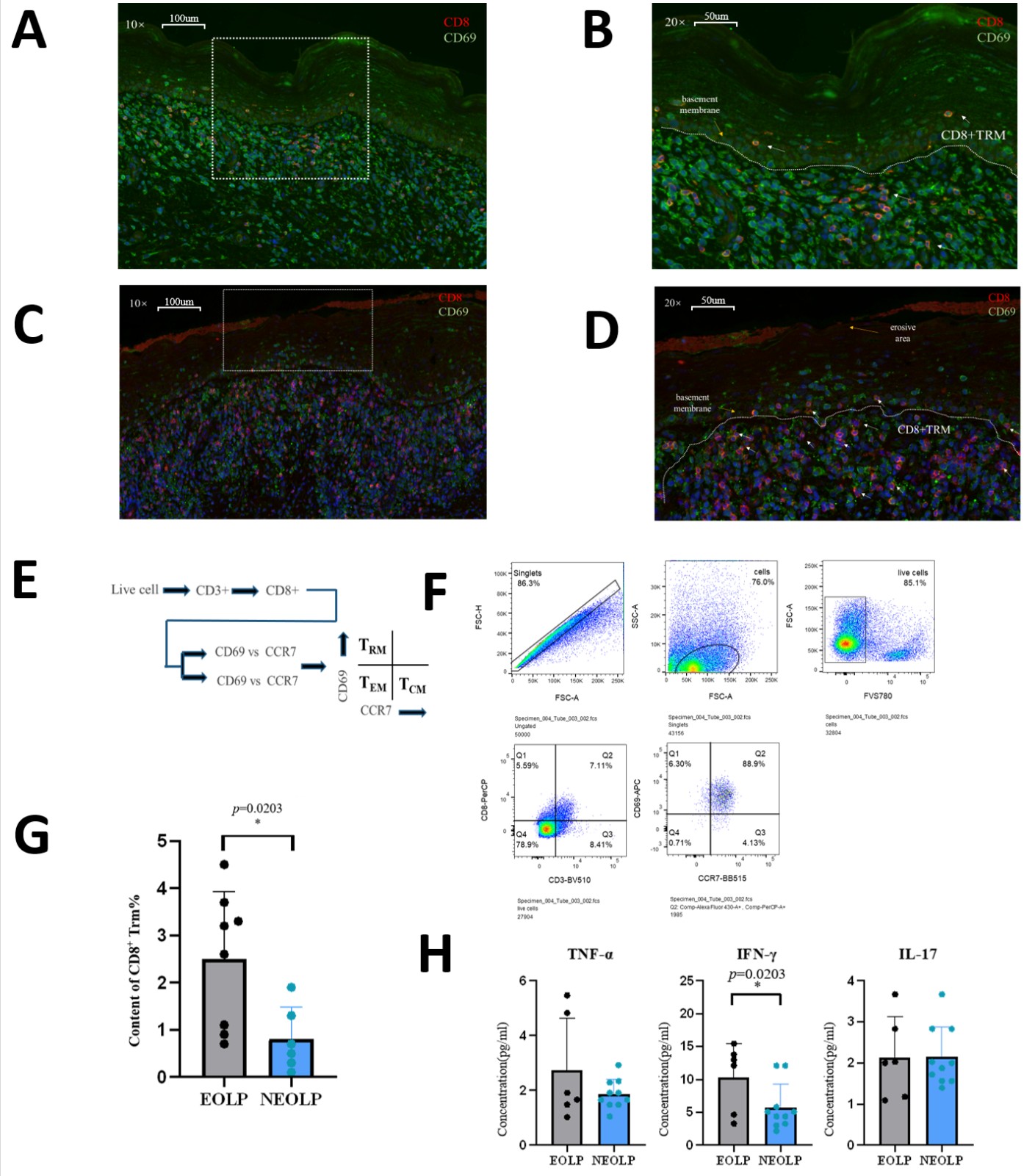

**Figure 5.** A high level of CD8+Trm in erosive oral lichen planus (EOLP) is closely related to the basement membrane and directly affects oral lichen planus's (OLP) clinical manifestations through active secretion. (**A, B**) Immunofluorescence images of CD8+ Trm cells in non-erosive oral lichen planus (NEOLP; the white dotted line is the basement membrane, and the red arrow points to the CD8+ TRM cells). (**C, D**) Immunofluorescence images of CD8+ Trm cells in EOLP (the white dotted line is the basement membrane, and the red arrow points to the CD8+ Trm cells). (**E**) Schematic diagram of CD8+ Trm

*Figure 5 continued on next page*

*Figure 5 continued*

sorting by flow cytometry. (**F**) Image of sorting CD8[+] Trm flow cytometry and gating strategy. (**G**) Comparison of the proportion of CD8[+] Trm in NEOLP versus EOLP. (**H**) Enzyme-linked immunosorbent assay (ELISA) detection results of CD8[+] TRM secreting TNF-α, IFN-γ, and IL17 in NEOLP versus EOLP.

The online version of this article includes the following source data for figure 5:

**Source data 1.** CD8+ Trm content in EOLP and NEOLP by flow cytometry.

**Source data 2.** ELIASA detection of CD8+ Trm secreted TNF-α, IFN-γ and IL17 contents in EOLP and NEOLP.

showed that CD8[+] Trm cells in EOLP produced significantly higher levels of the inflammatory cytokines TNF-α, IFN-γ, and IL17 than those in NEOLP, with the increase in IFN-γ being statistically significant (*Figure 5H*). It also confirmed the previous experimental results and indicated that CD8[+] Trm cells may affect the clinical manifestations of OLP through the secretion of the inflammatory factor IFN-γ.

## Discussion

This study is the first to provide a comprehensive characterization of the presence, spatial distribution, and heterogeneity of CD8[+] Trm cells in OLP tissues with different clinical manifestations, revealing possible regulatory mechanisms of CD8[+] Trm cells in OLP pathology. Our findings suggest that CD8[+] Trm cells play a crucial role in the pathological process of OLP and contribute to the erosion of OLP by secreting multiple cytokines.

We utilized scRNA-seq combined with spatial transcriptome to summarize the immune microenvironment of normal oral mucosa and OLP with different clinical subtypes. This study displayed that CD8[+] Trm cells exist in OLP and normal tissues and the content of CD8[+] Trm cells in normal mucosa, NEOLP, and EOLP increased gradually. Compared with NEOLP, CD8[+] Trm cells in EOLP were more extensive, especially in the epithelial deletion area. Pseudo-time analysis revealed that CD8[+] Trm cells represent one of the terminal states of T cells in OLP.

CD8[+] Trm cells may affect the biological process of OLP by releasing cytokines such as TNF, IFN, IL17, and the activity of CD8[+] Trm cells was enhanced in EOLP. It is suggested that the presence of CD8[+] Trm cells may be strongly correlated with the clinical type of OLP. Previous studies have found that CD8[+] Trm cells can produce IFN-γ, which is a key cytokine in response to viral infection (*Cheuk et al., 2017*; *Guidotti et al., 1996*). CD8[+] Trms can also produce perforin and granzyme B upon IL15 stimulation to mediate cytotoxic responses that respond to infections by enhancing local immune responses (*Cheuk et al., 2017*). In addition, IFN-γ upregulates the production of vascular cell adhesion molecule-1, which recruits central memory T cells (TCM) and B cells to the site of Trm cell localization (*Ariotti et al., 2014*).

Subsequently, we constructed an OLP clinical follow-up cohort and performed bulk RNA-seq of local lesions in patients to further investigate the role of CD8[+] Trm cells in OLP progression. We found that compared with NEOLP, CD8[+] Trm cells marker genes and related factors were significantly increased in EOLP. Moreover, follow-up analysis revealed that the recurrent erosion group had significantly higher expression levels of CD8[+] Trm cell marker genes and related immune factors than the persistent non-erosion group, consistent with the EOLP results, which validated the accuracy and consistency of our findings. Multiple regression analyses based on OLP clinical cohort showed that multiple core genes of CD8[+] Trm were closely related to OLP outcomes, which may be a significant factor leading to the deteriorative clinical outcome of OLP by causing repeated erosive lesions through local activation. Similarly, CD8[+] Trm cells also play a crucial part in the recurring aggravation of psoriasis. Even after the psoriatic skin lesions have disappeared, CD8[+] Trm cells can still be detected in the skin that seems normal. These cells have the potential to escalate local inflammation over time, trigger recurrent attacks at the same site, and generate a domino effect of inflammation (*Owczarczyk Saczonek et al., 2020*).

Moreover, immunofluorescence and flow cytometry experiments confirmed that the distribution of CD8[+] Trm cells was closely related to the erosion of epithelial tissue, and its content in EOLP was significantly higher than that in NEOLP. An ELISA test confirmed that CD8[+] Trm cells in EOLP tissue have enhanced ability to produce inflammatory cytokines such as IFN-γ compared with NEOLP, which may be an important promoting factor of epithelial erosion. Previous evidence suggests that CD8[+] Trm cells protect the host against viruses by producing IFN-γ in lung tissue (*Jiang et al., 2022a*).

Immune and inflammatory factors are considered to play an important role in the development of OLP, and cytokines IFN-γ may contribute greatly to the immune pathogenesis of OLP.

Recurrent episodes of autoimmune diseases are believed to be due, in part, to the persistence of local Trm cells that are difficult to eliminate entirely (*Jiang et al., 2012*; *Owczarczyk Saczonek et al., 2020*). Immunosuppressive medications and local, or systemic corticosteroids are the main treatments for symptomatic OLP (*García-Pola et al., 2017*; *Raj et al., 2021*). We discovered that there was no statistically significant difference between medication therapy and repeated degradation of OLP in our cohort analysis. This might be because CD8+ Trm is difficult for the present medications to entirely eradicate. Therefore, through a series of studies, we have confirmed that CD8+ Trm cells are not only increased in number but also functionally active in EOLP. These cells are primarily distributed in the lamina propria close to the basement membrane and release a variety of inflammatory factors. The release of excess cytokines such as IFN-γ may influence the clinical types and outcomes of OLP, making CD8+ Trm cells an important promoting factor for the recurrent erosion of OLP. CD8+ Trm cells may be a great important promoting factor for the recurrent erosion of OLP, which may become a potential immunotherapy target and provide new ideas for the treatment of OLP.

Our study has two limitations. Firstly, scRNA-seq was not performed on normal oral mucosal tissue to explore the role of CD8+ Trm cells in the pathogenesis of OLP. Secondly, since there is currently no animal model for OLP, a cell co-culture model involving CD8+ Trm cells could provide more evidence in further studies. Despite these limitations, our findings support that CD8+ Trm cells play a considerable role in the disease process of OLP and a critical factor in the worsening of its clinical manifestations.

## Materials and methods
### Human samples
All individuals provided written informed consent and this study was supported by the Ethics Committee of West China Hospital of Stomatology Sichuan University [WCHSIRB-2019-167]. According to the lesion with or without erosion in the biopsy, this study classified OLP as two groups, NEOLP and OLP.

### Cell isolation and processing
In the framework of the study, OLP tissues were either processed immediately for scRNA-seq or flow cytometry and cell sorting and stored at −80°C refrigerator in freezing for bulk RNA-seq. All biopsied mucosa samples were immediately processed for scRNA-seq or flow cytometry and cell sorting. Samples were gently removed adipose tissue and minced with scissors in a sterile tissue culture dish.

Tissue fragments were then digested with enzyme mixture 500 µl (whole-skin dissociation kit, Miltenyi Biotec) in gentleMACS C tubes (Miltenyi Biotec) which were incubated in a 37°C water bath for 3 hr under manual agitations every 15 min. Samples were mechanically separated with gentleMACS Dissociator (Miltenyi Biotec) for 1 min and then spun to collect.

Then move C tubes back to the water bath for another 20 min. Next, enzymes were inactivated with 1 ml precool DMES (Dulbecco's Modified Eagle Medium with Serum) from Gibco Laboratories at the end of the incubation. Samples were filtered through a 70 µm strainer (BD Bioscience) and collected by centrifuging in a table-top centrifuge at 400 × $g$ at 4°C for 5 min. Samples were treated with 5 ml ACK lysis buffer (Ammonium-Chloride-Potassium lysis buffer, Miltenyi Biotec) for 5 min and centrifuged at 400 × $g$ at 4°C for 5 min. Remove the supernatant, and added 100 µl dead cell removal kit (Miltenyi Biotec) for 15 min. Then added 1500 µl into the cell suspension three times, and through the magnetic column (Miltenyi Biotec) after pipetting and mixing. Samples were centrifuged at 400 × $g$ at 4°C for 5 min and resuspended in 200 µl Dulbecco's Phosphate Buffered Saline (D-PBS).

### 10× Genomics library preparation and sequencing
The 10×Genomics Chromium System is a microfluidic platform based on Gel-bead in EMulsion (GEM) technology for generating real test datasets.

Gel beads containing barcode information were combined with a mixture of cells and enzymes and then encapsulated in microfluidic droplets to form GEMs.

The GEMs flowed into a reservoir and were collected, the gel beads were lysed to release the barcode sequences, the cDNA fragment was reverse transcribed, and samples were labeled. The gel beads were broken and the oil droplets were broken up and PCR amplification was performed using

the cDNA as a template. The products of all GEMs are mixed and a standard sequencing library was constructed after using the Chromium Single Cell 5′ library or 3′ v2 library preparation kit according to the manufacturer's protocol (10× Genomics). Finally, all sequencing experiments were conducted using Illumina NextSeq 500 in the Genomics Sequencing.

## Cell clustering and cell-type annotation

The R package Seurat (v 4.0.1) was used to cluster the cells in the merged matrix. The first step was to filter out low-quality cells, which included cells with less than 500 transcripts, less than 100 genes, or cells with more than 10% of mitochondrial expression. And then normalized the data used the NormalizeData function. The canonical correlation analysis was performed using the normalized expression levels, the batch effect was corrected, and the data were integrated. *Z*-score normalization is performed on the integrated data, and principal component analysis (PCA) is performed using the normalized expression. The goal of PCA is to reduce the dimension of feature vectors by compressing the scale of the original data matrix, representing the most important features with the least dimension, and the new variable is a linear combination of the original variables, reflecting the comprehensive effect of the original variables. Dimensionality reduction through PCA reduced variables and finally clustered the cells.

## Cell-type subclustering and cell trajectory analysis

The prevalent cell types underwent subclustering. The subclusters were obtained using the identical functions as previously mentioned. We eliminated from further analysis subclusters that were solely defined by mitochondrial gene expression, a sign of low quality. By crossing over the canonical subtype signature genes with the marker genes for the subclusters, the subtypes were annotated. To identify the canonical pathways and putative upstream regulators, IPA was applied to the DEGs. Significant upstream regulators were those with activation *z*--scores ≥2 or ≤2. Using the Add Module Score function on the genes activated by the targeted cytokine from bulk RNA-seq data, as previously stated, the module scores were generated.

In this study, Monocle 2 was used for cell trajectory analysis. First, all DEGs in cell subtypes (clusters) were screened; then dimensionality reduction was performed and then a minimum spanning tree was constructed; Finally, the best cell development or differentiation pseudo-time trajectory curve was fitted.

## ST process

The ST protocol was performed according to recommendations (10× Genomics). Fresh OLP tissues were put into a mold in powdered dry ice, wrapped with optimal cutting temperature compound (OCT), completely frozen and optimized to ideal form, and then sectioned while Hematoxylin and Eosin (HE) stained. The OCT-embedded tissues were cut to 10 µm thickness with a cryostat, then adhered to the chips, and frozen sectioned.

The chips with the tissue section were fixed with methanol, then stained with HE, and incubated with permeabilase on a PCR adapter to release the mRNA in the cells and bind to the corresponding capture probe. Validation of chips with brightfield imaging and fluorescence imaging. And then cDNA synthesis and sequencing library preparation using captured RNA as template.

Then, the prepared sequencing library is subjected to second-generation high-throughput short-read sequencing; finally, combined with the HE results, the expression of genes, the level of expression, and the spatial location information of these genes are determined.

## ST analysis

Samples' data were initially processed using the 10× official software Space Ranger (10× Genomics). Space Ranger displays the captured area organized in the chip through an image processing algorithm and distinguished the reads of each spot according to the spatial barcode information. The number of pair reads, the numbers of detected genes, and the number of UMIs in each spot were counted to evaluate the quality of the samples. The data were then normalized using sctransform to construct a regularized negative binomial model of gene expression to detect high variance features. After overall quality control, the proportion of mitochondrial genes in each sample is less than 0.12%,

and the overall average is less than 0.08%, which meets the requirements of the 10× Genomics Visium platform.

After further dimensionality reduction, the gene expression of each spot was used to cluster the same type of spots to form spots clusters. Finally, through the labeling of characteristic genes, transcriptome visualization in the tissue space can be realized.

## Immunofluorescence staining

5 μm formalin-fixed mucosa lesion biopsy tissue sections were dewaxed in xylene and rehydrated with distilled water. Membrane rupture after proteinase K repair, then, the terminal deoxynucleotidyl transferase (TDT) enzyme, deoxyuridine triphosphate (dUTP) and buffer in the tunel kit were mixed at a ratio of 1:5:50, added to the tissue area, and then transferred to the wet box, and incubated a 37°C for 2 hr. After elution, the tissue surface was covered with 3% bovine serum albumin (BSA) and blocked at room temperature for 30 min. Then, incubation with the following primary monoclonal antibodies was performed: rabbit anti-human CD8, and mouse anti-human CD69, all at 1:200 dilution.

## Flow cytometry and cell sorting

Surface marker staining after resuspending the digested single-cell suspension from the lesion of mucosa in 1–2 ml of BD Pharmingen Stain Buffer (BSA) pre-chilled at 4°C.

After rinsing twice with BSA, the cells were stored at 4°C in the dark, and flow cytometry was performed immediately. The BD cytometer was used and samples were sorted with fluorescence-activated cells, and the scheme of surface staining was as indicated in the text, using 4',6-diamidino-2-phenylindole (DAPI) as the reactive dye.

## ELISA procedure

After sorting the CD8[+] TRM cells, add 10 μg/ml PHA to its 1640 medium. The supernatant was collected after culturing at 5% $CO_2$ and 37°C for 72 hr, and then an ELISA assay was performed to detect TNF-α, IFN-γ, and IL17, respectively.

Add deionized water to the corresponding reagents in the kit to prepare the detergent, diluent, and standard solution, prepare a standard dilution tube and prepare the color developer 15 min before use. Then, standard substances and experimental samples of different concentrations were added to the corresponding microwells, and 100 μl was added to each microtiter well and incubated for 2 hr after sealing.

After washed of each well, added 200 μl of enzyme-labeled detection antibody, then sealed the plate and incubated for 2 hr before washing, added 200 μl of a diluted chromogenic substrate, sealed the plate and incubated for 30 min, and added 50 μl of stop solution to each well. Measured the absorbance at 450 nm with a microplate reader, set 540 nm as the calibration wavelength, and measured the OD value of each well after zero-adjusting the blank control well. Finally, made a standard curve according to the concentration and OD value of the standard product, and then calculated the sample concentration according to the standard curve equation.

## Bulk-RNA sequencing procedure

Sample RNA was extracted by Cetyl Trimethyl Ammonium Bromide (CTAB) method (*Wang and Stegemann, 2010*). And the extracted RNA was then tested for purity, concentration, and integrity. A number of library construction procedures, including end repair, end addition of A, ligation adapter addition, and fragment security screening, were carried out once mRNA is recovered. It was sequenced on the device following PCR amplification and purification.

## Clinical cohort and transcriptome data analysis

*T*-tests for continuous variables and chi-square tests for categorical variables were used to analyze the baseline data of the clinical cohort's patients. *T*-tests were also used to determine whether one gene's expression varied from that of another.

Multivariate logistic regression was used to analyze the effects of related genetic factors on the clinical outcome of OLP to account for confounding. Analysis based on a variety of variables. R software (Version 4.0.1) was used for the analysis and statistics of this portion of the statistical data, and all of the aforementioned analysis and test levels were set to 0.05.

Unreliable data were filtered and eliminated by the quality assessment of the original data after sequencing is complete, and downstream analysis was carried out following gene difference analysis. Correlation analysis, PCA, sample cluster analysis, and weighted gene co-expression network analysis had next be performed.

## Acknowledgements

This work was supported by grants from the National Natural Science Foundation of China (No. 81730030 and No. 82001059). Thanks for the support of the department of oral medicine of West China Hospital of Stomatology. Thanks to all clinical participants for their contribution.

## Additional information

### Funding

| Funder | Grant reference number | Author |
| --- | --- | --- |
| National Natural Science Foundation of China | 81730030 | Qianming Chen Hao Xu |

The funders had no role in study design, data collection, and interpretation, or the decision to submit the work for publication.

### Author contributions

Maofeng Qing, Conceptualization, Formal analysis, Validation, Writing – original draft; Dan Yang, Formal analysis, Writing – review and editing; Qianhui Shang, Validation, Visualization, Writing – original draft; Jiakuan Peng, Software, Formal analysis, Methodology, Writing – review and editing; Jiaxin Deng, Data curation, Formal analysis, Writing – review and editing; Jiang Lu, Jing Li, Yu Zhou, Supervision, Project administration, Writing – review and editing; HongXia Dan, Resources, Supervision, Writing – review and editing; Hao Xu, Conceptualization, Resources, Formal analysis, Funding acquisition, Project administration, Writing – review and editing; Qianming Chen, Conceptualization, Funding acquisition, Project administration, Writing – review and editing

### Author ORCIDs

Maofeng Qing https://orcid.org/0000-0001-7829-5564
Hao Xu http://orcid.org/0000-0002-5665-0139
Qianming Chen http://orcid.org/0000-0002-5371-4432

### Ethics

All individuals provided written informed consent and this study was supported by the Ethics Committee of West China Hospital of Stomatology Sichuan University [WCHSIRB-2019-167].

### Decision letter and Author response

Decision letter https://doi.org/10.7554/eLife.83981.sa1
Author response https://doi.org/10.7554/eLife.83981.sa2

## Additional files

### Supplementary files

• Supplementary file 1. Gene lists and quantification data for OLP and CD8+ Trm subgroups, and spot numbers of each sample in ST.

• MDAR checklist

### Data availability

The data of this study, including scRNA-seq data, ST data, and bulk RNA-seq data are available in the Gene Expression Omnibus (GEO) database, accession numbers GSE213345, GSE213346, and GSE211630.

The following datasets were generated:

| Author(s) | Year | Dataset title | Dataset URL | Database and Identifier |
|---|---|---|---|---|
| Shang Q, Qing M, Xu H, Chen Q | 2023 | Immune microenvironment and molecular mechanism of oral lichen planus [Spatial transcriptomics] | https://www.ncbi.nlm.nih.gov/geo/query/acc.cgi?acc=GSE213345 | NCBI Gene Expression Omnibus, GSE213345 |
| Qing M, Shang Q, Xu H, Chen Q | 2023 | Immune microenvironment and molecular mechanism of oral lichen planus [bullk RNA-Seq] | https://www.ncbi.nlm.nih.gov/geo/query/acc.cgi?acc=GSE213346 | NCBI Gene Expression Omnibus, GSE213346 |
| Shang Q, Xu H, Chen Q | 2023 | Immune microenvironment and molecular mechanism of oral lichen planus [Single cell RNA-Seq] | https://www.ncbi.nlm.nih.gov/geo/query/acc.cgi?acc=GSE211630 | NCBI Gene Expression Omnibus, GSE211630 |

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
