## [Editor Report]

Overall, this manuscript is a valuable contribution to the field of human immunology and provides solid data and interpretations. Very little is known about oral lichen planus so this dataset may also serve as a public resource. Thank you for your contribution on mapping the cellular landscape of this poorly understood condition.

---

## [Decision Letter]

**Decision letter after peer review:**

Thank you for submitting your article "CD8^+^ tissue-resident memory T cells triggered the erosion of oral lichen planus by the cytokine network" for consideration by *eLife*. Your article has been reviewed by 2 peer reviewers, including Brian S Kim as Reviewing Editor and Reviewer #2, and the evaluation has been overseen by Tadatsugu Taniguchi as the Senior Editor.

Essential revisions:

1) Please temper claims about CD8 Trms being drivers of disease as this is not proven.

2) Please clarify why erosive vs. non-erosive is a critical distinction in light of data suggesting it is more of a spectrum.

3) Please clarify the source of IL-23 as its unconventional interpretations of IL-23 biology are mentioned.

4) Please analyze figures and add clarifying notations as requested by both reviewers.

*Reviewer #1 (Recommendations for the authors):*

1) Need n>1 per group when comparing groups (EOLP).

2) I don't understand the value of comparing NEOLP and EOLP lesions. EOLP is likely to be a more severe presentation of the same disease. Perhaps the data for NEOLP and EOLP could be pooled and compared with healthy oral mucosa to yield greater insight into OLP pathogenesis.

3) Figures 1 and 2 could be improved by adding some additional or expanded displays of scRNA-seq data. Including the top 10-50 DEGs/cluster and specificity controls (non-T cells). Also, a much more detailed description of methods and manner in which data is displayed is needed in Figure 2 legend and methods section as in comment #1. As an example, for Figure 2 F, I am not clear on what is meant by 'gene ratio' and whether data are displayed as -log P values. Values of 0.05-0.15 seem very low to me (background).

4) Why focus on Runx1 and il23a in Figure 3? Were these highlighted bc they are among the most highly expressed genes? Please include a more comprehensive analysis of spatial gene expression in OLP and healthy mucosa as this is very interesting data. Also, it would be nice to have some higher-power images of select areas for ease of interpretation.

5) Please comment on whether you think IL-23 protein is produced by CD8 Trm, as T cells have not previously been reported to make IL-23. Is il12b co-expressed by cells in which you see il23a signal?

6) Please explain what 'differential gene analysis' was done in the cohort study described in Tables 1, 3, and FiguresS2A-B.

7) I am intrigued by the logistic regression analyses described in Table 3. However, it is unclear to me what the impact of each indicated gene is on the clinical type and clinical outcome. In the Results section text, the authors state that "CD69, Itga1, LITAF, SKIL, and cytokine-related genes such as… Were associated with two clinical outcomes of persistent non-erosion and recurrent erosion." Can the authors give more detail on how they reached this conclusion and how the data in Table 3 can be used to describe an association with a specific outcome or clinical type?

8) Gating strategy is unclear, one cannot interpret these data without them. Please label all plot axes and gated populations. What gates or populations do the data in Figure 4H derive from?

9) In all bar graphs please plot individual values as well as mean plus error bars.

10) The manuscript is difficult to read with a number of semantic errors including in the title. Please revise appropriately.

*Reviewer #2 (Recommendations for the authors):*

– Comment on the role(s) of B cells and mast cells in OLP pathogenesis since they are heavily represented specifically in erosive OLP tissue.

– Revise the text to emphasize CD8^+^ Trm cells as a defining feature of erosive OLP rather than a mechanistic driver of disease.

– Assess proliferative capacity and/or activation state of primary CD8^+^ Trm. Quantitative PCR analysis could also provide robust measures of cytokine production.

Clearly label Figure 4 flow cytometry data; replace fluorochromes with markers used and label axes with units of measurement that were used.

---

## [Author Response]

Essential revisions:1) Please temper claims about CD8 Trms being drivers of disease as this is not proven.

Thanks for pointing it out, and sorry for not being very clear in our previous article. Indeed, CD8^+^ TRM was found in our study to be an important cause of changes in the local presentation of OLP (erosions), rather than a clear driver of disease; this has been revised and emphasized in the text.

“The purpose of writing in the introduction part of the article is changed as follows:

CD8^+^ Trm cells are considerable components of local immunity, yet their presence, distribution, and function in OLP are poorly understood. This study aims to investigate the presence and spatial distribution of CD8^+^ Trm cells in different clinical manifestations of OLP, and to determine their functional role, especially in the context of the heterogeneity observed between NEOLP and EOLP. Additionally, the study aims to explore the underlying molecular mechanisms that contribute to the development of erosive lesions in OLP.” (Page 5, lines 8-14)

And the first paragraph of the Discussion section is then amended to read as follows:

“This study is the first to describe provide a comprehensive characterization of the presence, spatial distribution, and heterogeneity of CD8^+^ Trm cells in OLP tissues in with different clinical manifestations, revealing possible regulatory mechanisms of CD8^+^ Trm cells in OLP pathology. and to explore the regulatory mechanism of CD8^+^ Trm cells in OLP pathology. Our findings suggest that CD8^+^ Trm cells play a crucial role in the pathological process of OLP and contribute to the erosion of OLP by secreting multiple cytokines.” (Page 12, lines 4-10)

2) Please clarify why erosive vs. non-erosive is a critical distinction in light of data suggesting it is more of a spectrum.

Thank you for your suggestion. Erosion is an important factor affecting the quality of life of patients and the frequency of visits to the doctor. OLP with repeated erosion is more likely to develop cancer, and OLP cannot be cured. Therefore, it is particularly important to minimize the occurrence of erosion in clinical practice. We have revised and emphasized in the text.

The modified description in introduction is as follows:

“Notably, EOLP has a significantly higher risk of malignant transformation than non-erosive oral lichen planus (NEOLP) (Danielsson et al., 2013). To reduce the psychological and economic burden of OLP patients, improve their quality of life, and decrease the risk of cancer, it is crucial to maintain the disease in a relatively stable non-erosive stage for as long as possible. However, clinical experience suggests that OLP often exhibits a prolonged and recurrent disease course, with alternating periods of non-erosive and erosive lesions. Despite this, the underlying causes and mechanisms of lesion type switching remain unclear (Husein-ElAhmed and Steinhoff, 2022).” (Page 4, lines 13-21)

3) Please clarify the source of IL-23 as its unconventional interpretations of IL-23 biology are mentioned.

Thank you for pointing out our mistake. We made a mistake in the article writing. In our study, IL23A was not produced by T cells, but was significantly increased in erosive OLP, and IL-23A has the function of to receptors to activate memory T cells, so we analyzed that this may be one of the reasons for the increased Trm in erosive OLP, which has been modified in the text.

The reviewer pointed out that our erosive OLP sample was relatively weak in the single-cell part, so we added another case of erosive OLP later. After completely reanalyzing the single-cell part, the expression of IL23A in NEOLP and EOLP. The difference is not as good as other factors, The results of the redone analysis no longer involve IL23A, so we deleted it in the text and rewrote this part.

The modified description in this part is as follows:

“Further analysis of the significantly different genes between NEOLP and EOLP indicated that B2M, closely related to the stability of MHC class I molecules and antigen presentation, and HIF1A, which regulates immune cell function and inflammation, were found to be significantly expressed in EOLP. Moreover, the EOLP expressed a number of markers involved in differentiation and function of Trm cells, including CD69, IL7R, CD7, FOS, etc. (Figure S3A).” (Page 7, lines 23-28)

4) Please analyze figures and add clarifying notations as requested by both reviewers.

We have made corresponding modifications and additions according to the reviewers' comments.

Reviewer #1 (Recommendations for the authors):1) Need n>1 per group when comparing groups (EOLP).

Thank you for your valuable comments, let us also realize that the number of EOLP is only one case is lacking. We added the single-cell data of a patient with EOLP, and re-did the results of this part of the single-cell analysis. There is a good improvement in the quality of the article itself.

These include the first three sections of the results part.

“The number of cells contained in the final data set increased from 31,330 to 46,377, and the average number of genes per cell increased from 1470 to 1743. Visualization using Unified Manifold Approximation and Projection (UMAP) showed a rise from 28 to 47 cell clusters annotated with 8 dominant cell types. Compared with previous studies, with the addition of new samples, our research conclusions have not changed significantly.”

2) I don't understand the value of comparing NEOLP and EOLP lesions. EOLP is likely to be a more severe presentation of the same disease. Perhaps the data for NEOLP and EOLP could be pooled and compared with healthy oral mucosa to yield greater insight into OLP pathogenesis.

Thank you for your suggestion. OLP is a very complex immune-related disease, and its pathogenesis is very complicated. Therefore, there is no corresponding animal model at present. The focus of this study is not the cause of the disease, but the relationship between different clinical manifestations of the disease. At the same time, there are ethical issues, healthy people are not convenient to access their oral tissue. Therefore, this study did not involve normal tissue research controls. Due to the complex etiology of OLP, it may be the result of a multi-factor effect, and currently there is no cure. Erosion is an important factor affecting the quality of life of patients and the frequency of visits to the doctor, and OLP with repeated erosion is more likely to develop cancer, and OLP cannot be cured, so minimizing the occurrence of erosion is particularly important in clinical practice. In response to these problems, we have made modifications and clarifications in the article.

The modified description in introduction is as follows:

“Notably, EOLP has a significantly higher risk of malignant transformation than non-erosive oral lichen planus (NEOLP) (Danielsson et al., 2013). To reduce the psychological and economic burden of OLP patients, improve their quality of life, and decrease the risk of cancer, it is crucial to maintain the disease in a relatively stable non-erosive stage for as long as possible. However, clinical experience suggests that OLP often exhibits a prolonged and recurrent disease course, with alternating periods of non-erosive and erosive lesions. Despite this, the underlying causes and mechanisms of lesion type switching remain unclear (Husein-ElAhmed and Steinhoff, 2022).” (Page 4, lines 13-21)

3) Figures 1 and 2 could be improved by adding some additional or expanded displays of scRNA-seq data. Including the top 10-50 DEGs/cluster and specificity controls (non-T cells). Also, a much more detailed description of methods and manner in which data is displayed is needed in Figure 2 legend and methods section as in comment #1. As an example, for Figure 2 F, I am not clear on what is meant by 'gene ratio' and whether data are displayed as -log P values. Values of 0.05-0.15 seem very low to me (background).

Thank you for your suggestion. We reorganized and rewritten the single-cell data, which is specifically reflected in the figures and supplementary figures and tables involved in the first three parts of the Results section of the article.

4) Why focus on Runx1 and il23a in Figure 3? Were these highlighted bc they are among the most highly expressed genes? Please include a more comprehensive analysis of spatial gene expression in OLP and healthy mucosa as this is very interesting data. Also, it would be nice to have some higher-power images of select areas for ease of interpretation.

In the previous single-cell analysis, RUNX-1, IL23A were both significantly elevated in EOLP, so we conducted further analysis on them. But after adding new EOLP samples, our single-cell data changed, RUNX-1, IL23A decreased significantly, while B2M, HIF1A, and Trm-related factors such as CD69, IL7R, CD7, etc. were very significant in EOLP obvious.

5) Please comment on whether you think IL-23 protein is produced by CD8 Trm, as T cells have not previously been reported to make IL-23. Is il12b co-expressed by cells in which you see il23a signal?

Thank you for pointing out our mistake. We made a mistake in the article writing. In our study, IL23A was not produced by T cells, but was significantly increased in erosive OLP, and IL-23A and receptor When combined, with the function of memory T cell activation, we analyzed that this may be one of the reasons for the increased TRM in erosive OLP. With the addition of new EOLP samples, our single-cell data changed, and the significance of IL23A decreased, while B2M, HIF1A, and Trm-related factors such as CD69, IL7R, and CD7 were significantly significant in EOLP.

We modified the section as follows:

“Further analysis of the significantly different genes between NEOLP and EOLP indicated that B2M, closely related to the stability of MHC class I molecules and antigen presentation, and HIF1A, which regulates immune cell function and inflammation, were found to be significantly expressed in EOLP. Moreover, the EOLP expressed a number of markers involved in differentiation and function of Trm cells, including CD69, IL7R, CD7, FOS, etc. (Figure S3A).” (Page 7, lines 23-28)

6) Please explain what 'differential gene analysis' was done in the cohort study described in Tables 1, 3, and Figures S2A-B.

Thank you for your comments. In this part, the difference analysis of CD8^+^ TRM core genes and inflammatory factor genes is carried out. Table 1 does not involve the analysis of differential genes, it’s analysis of clinical variables. And in Table 3, and Figure S2 we did logistic regression analysis of CD8^+^ Trm core transcriptome in different OLP clinical types and clinical outcomes.

The modified description in introduction is as follows:

“Moreover, the multiple core factors and effector cytokines of CD8^+^ Trm cells were found to be consistently upregulated in the recurrent erosion group, further supporting the role of CD8^+^ Trm cells in OLP progression (Figure S4C and S4D). To mitigate confounding factors and minimize systematic bias, we employed multivariate logistic regression to analyze patients with different clinical types and outcomes of OLP.” (Page 10, lines 14-20)

7) I am intrigued by the logistic regression analyses described in Table 3. However, it is unclear to me what the impact of each indicated gene is on the clinical type and clinical outcome. In the Results section text, the authors state that "CD69, Itga1, LITAF, SKIL, and cytokine-related genes such as… Were associated with two clinical outcomes of persistent non-erosion and recurrent erosion." Can the authors give more detail on how they reached this conclusion and how the data in Table 3 can be used to describe an association with a specific outcome or clinical type?

Our study correlated the expression of the core genes of CD8^+^ TRM with the clinical diagnoses of OLP and the clinical outcomes of the cohort follow-up using regression analysis to exclude other interfering factors.

After modified and tweaked, the section as following:

“Through differential gene analysis, it was found that in NEOLP and EOLP, the expression of multiple core factors and effector cytokines or their receptor transcription factors of CD8^+^ Trm cells, including ITGA1, LITAF, SKIL, etc., and related immune factor receptors such as TNFRSF6B, IL17RA, IFNAR1, etc., were significantly increased in the EOLP than in the NEOLP (Figure S4A and S4B; Table 3). To further investigate the clinical relevance of these findings, we divided the clinical cohort into two groups based on the 1-year follow-up: the recurrent erosion (RE) group (the interval between erosions <3 months) and the persistent non-erosion (PNE) group (≥3 months without any form of erosion). In addition to the above clinical factors being included in the two groups for statistical comparison, the diagnosis (NEOLP or EOLP) and medication (divided into three groups: local glucocorticoids, local glucocorticoids + immunosuppressant, and other drugs) were analyzed (Table 2), in which glucocorticoids were all used for local and external use. Differences in clinical information other than diagnosis were still not statistically significant for the clinical outcome of OLP. Moreover, the multiple core factors and effector cytokines of CD8^+^ Trm cells were found to be consistently upregulated in the recurrent erosion group, further supporting the role of CD8^+^ Trm cells in OLP progression (Figure S4C and S4D).

To mitigate confounding factors and minimize systematic bias, we employed multivariate logistic regression to analyze patients with different clinical types and outcomes of OLP. The multivariate logistic model was used to perform regression analysis on the effects of CD8^+^ Trm-related factors on the clinical manifestations and clinical outcomes of OLP after adjusting for clinical factors such as age, gender, smoking, and drinking. We found that the core genes of CD8^+^ Trm cells are closely related to the clinical manifestations of OLP, and the expression differences of marker genes such as ITGA1, LITAF, SKIL, and cytokine-related genes such as IL17RA, IFNG41, IFNAR1, and TNFRSF6B were statistically significant between the two clinical types. The core genes of CD8^+^ Trm cells are also closely related to the erosion of OLP. Marker genes such as CD69, ITGA1, LITAF, SKIL, and cytokine-related genes such as IL17RA, IFNG41, IFNAR1, and GZMB are associated with two clinical outcomes of persistent non-erosion and recurrent erosion. The differences in expression were statistically significant and were significantly increased in the recurrent erosion group (Table 3).” (Page 9, lines 33-Page 11, lines 3)

8) Gating strategy is unclear, one cannot interpret these data without them. Please label all plot axes and gated populations. What gates or populations do the data in Figure 4H derive from?

Thanks for your suggestion, your questions are valuable. We got strategy for CD8^+^ TRM derived from related literature. (Bishu S, El Zaatari M, Hayashi A, Hou G, Bowers N, Kinnucan J, Manoogian B, Muza-Moons M, Zhang M, Grasberger H, Bourque C, Zou W, Higgins PDR, Spence JR, Stidham RW, Kamada N, Kao JY. CD4^+^ Tissue-resident Memory T Cells Expand and Are a Major Source of Mucosal Tumour Necrosis Factor α in Active Crohn's Disease. J Crohns Colitis. 2019 Jul 25;13(7):905-915.) Figure has been modified.

9) In all bar graphs please plot individual values as well as mean plus error bars.

Thank you for your valuable comments, we have modified the bar chart according to your comments.

10) The manuscript is difficult to read with a number of semantic errors including in the title. Please revise appropriately.

Thank you for your valuable feedback on our manuscript. Thank you for taking the time and effort to review our work.

We apologize for the difficulties you encountered in reading our manuscript and for the semantic errors in the title. We have thoroughly revised the manuscript to ensure its clarity and conciseness.